# The Effect of Surgical Masks on the Featural and Configural Processing of Emotions

**DOI:** 10.3390/ijerph19042420

**Published:** 2022-02-19

**Authors:** Natale Maiorana, Michelangelo Dini, Barbara Poletti, Sofia Tagini, Maria Rita Reitano, Gabriella Pravettoni, Alberto Priori, Roberta Ferrucci

**Affiliations:** 1Aldo Ravelli Research Center for Neurotechnology and Experimental Brain Therapeutics, Department of Health Science, University of Milan, Via A. di Rudinì, 8, 20142 Milan, Italy; natale.maiorana@unimi.it (N.M.); michelangelo.dini@unimi.it (M.D.); alberto.priori@unimi.it (A.P.); 2Department of Neurology and Laboratory of Neuroscience, IRCCS Istituto Auxologico Italiano, Piazzale Brescia, 20, 20149 Milan, Italy; b.poletti@auxologico.it (B.P.); tagini.sofia@gmail.com (S.T.); 3ASST Santi Paolo e Carlo, San Paolo University Hospital, Via A. di Rudinì, 8, 20142 Milan, Italy; mariella.reitano@gmail.com; 4Department of Oncology and Hematology-Oncology, University of Milan, Via Festa del Perdono, 7, 20122 Milan, Italy; gabriella.pravettoni@ieo.it; 5Psycho-Oncology Division, IRCCS-Istituto Europeo di Oncologia, Via G. Ripamonti, 435, 20141 Milan, Italy; 6Fondazione IRCCS Ca’ Granda Ospedale Maggiore Policlinico di Milano, Via F. Sforza, 35, 20122 Milan, Italy

**Keywords:** COVID-19, surgical mask, emotion recognition, alexithymia, face processing, featural processing, configural processing

## Abstract

From the start of the COVID-19 pandemic, the use of surgical masks became widespread. However, they occlude an important part of the face and make it difficult to decode and interpret other people’s emotions. To clarify the effect of surgical masks on configural and featural processing, participants completed a facial emotion recognition task to discriminate between happy, sad, angry, and neutral faces. Stimuli included fully visible faces, masked faces, and a cropped photo of the eyes or mouth region. Occlusion due to the surgical mask affects emotion recognition for sadness, anger, and neutral faces, although no significative differences were found in happiness recognition. Our findings suggest that happiness is recognized predominantly via featural processing.

## 1. Introduction

As the COVID-19 pandemic spreads, surgical masks became the main tool for fighting the disease. The use of surgical masks is of fundamental importance to protect individuals from infection [1]; however, it should be noted that surgical masks make it impossible to get a full view of people’s faces, therefore impairing the ability to read the facial emotions of others [2]. Surgical masks cover the lower part of the face, which is an important area for nonverbal communication of emotional states [3]. Understanding other people’s emotions is a fundamental ability for humans and a fundamental function at the basis of social interactions [4].

Ekman and colleagues (1992) [5] indicated six facial expressions reflecting basic emotions: sadness, happiness, anger, disgust, fear, and surprise [5]. These emotions appear to be universal and not linked to the individual’s culture. On this basis, the facial action coding system (FACS) was developed [6]. The FACS aims to create a taxonomy of facial expressions. In the FACS system, each anatomically possible facial expression is associated with specific movements of one or more muscles, which form the action units, and their possible temporal segments [6]. Each emotion can thus be categorized on the basis of specific movements. Different emotional expressions have one or more action units linked with specific parts of the face [7]. Therefore, different facial features carry specific information depending on the emotion that is observed [8]. With the aim to clarify how humans extract information about emotions from a seen face, numerous studies have focused on the featural and configural processing of faces [9]. Featural processing concerns the processing of information carried by specific face parts, for example the shape of the nose or the size of the mouth, while configural processing concerns the spatial relation between face parts, for example the distance between eyes and mouth [10,11]. Composite effect shows that emotion recognition is based mostly on configural rather than featural processing [9] and that configural processing is automatic and cannot be easily suppressed. Some studies argue that emotion processing is a two-step process [12], or that configural and featural processing contribute differentially to the recognition of different emotions [13]. In the process of a completely visible face, it is possible to rely on featural and configural information. When a masked face is viewed, perception of emotions relies only on featural processing of the upper face region, forcing individuals to process only featural information deriving from the upper part of the face. Experimental evidence suggests that happiness recognition relies more on featural processing while configural processing is strongly involved in sadness and anger recognition [13]. Therefore, it is important to consider how the disruption of configural processing and forcing people to process only part of the featural information due to a face mask can impact the recognition of specific emotions. It is possible that the emotional processing of a face does not rely on featural and configural processing but is instead based on different emotion-specific mechanisms and neural networks [14]. In a case study reported by Pegna and colleagues (2005) [15], a cortically blind patient with bilateral destruction of visual cortices and consequent cortical blindness could guess the emotional facial expression being displayed. At the same time, the patient could not guess other emotional stimuli such as pleasant or unpleasant scenes of sports, sex, violence, or mutilation. Functional magnetic resonance imaging (fMRI) showed the activation of the right amygdala during the processing of emotional faces. Evidence of the importance of face features derives from eye tracker studies which have shown that, during free exploration of faces with different emotional expressions, participants tended to look longer at the eyes region than at the mouth region [16,17,18]. Studies have also shown that there are different fixation patterns depending on the emotions being observed, with fixation on the upper part in recognizing fear and surprise and on the lower part in recognizing happiness [8]. Experimental evidence supports the importance of contextual factors in emotion perception. Kret and de Gelder (2012) [19] compared the effect of Islamic headdresses vs. scarfs and caps on emotion recognition, finding that fear and anger recognition were more accurate in women wearing the niqab than in women wearing cap and scarf; conversely, happiness recognition was more accurate in women wearing cap and scarf than in women wearing the niqab. Graham and Ritchie (2019) [20] found that sunglasses reduced the rating of trustworthiness in the observer and impaired identity recognition. Further evidence of effects due to minimal occlusion of the eye region comes from a study made by Kramer and Ritchie (2016) [21], where participants had more difficulty in matching identity when one of the two faces shown wore glasses. Although the eye region has played a predominant role in theories of emotion recognition, several studies have cleared the role of the mouth region in emotion recognition. Applying the Bubbles technique to emotion recognition, Blais and colleagues (2012) [22] found that humans rely more on the mouth than on the eyes area in both static and dynamic emotion recognition tasks. The authors proposed that the mouth region contains more discriminative motions between expressions compared to the eye region. 

To date, it is not clear whether emotion perception from a seen face is a phenomenon that relies purely on configural or featural processing of the face elements. Many findings show that emotion perception is a complex phenomenon depending also on contextual factors and on the specific emotion perceived by the observer. In the last two years many studies have focused their attention on the role played by surgical masks in the recognition of facial characteristics such as age, identity, or emotions, highlighting the impact of multiple fields of social cognition [2,23,24,25,26,27,28]. Surgical masks have a detrimental effect on matching face identity [28]. The specific mechanism underlying the effect of surgical masks on identity matching is due to qualitative and quantitative change in the way masked faces are perceived by the observer due to the lack of information carried out by the configural processing of a seen face [24].

Regarding the emotional process of a seen face, it was found that surgical masks superimposed on a portrait of a person mimicking the six basic emotions exert differential effects based on the emotion that was shown, with major effects on the recognition of angry, disgusted, and happy faces [23]. As reported by Marini and colleagues (2021) [29], surgical masks have different effects based on the specific emotion. In their study, subjects made more errors in sadness recognition than in happiness recognition when observing a masked face. This result is interesting, especially considering that the mouth region seems to be more informative in the recognition of positive emotions.

However, to date little is known about the specific mechanism that underlies these effects in terms of speed and accuracy. Some insight could be found in the study made by Fitousi and colleagues (2021) [30], where subjects were asked to discriminate between angry and neutral masked faces in the context of an inverted faces task. Results showed that the facial inversion effect was comparable in masked and unmasked conditions, demonstrating that emotion recognition with masked faces is based on featural rather than configural processing of the seen face. It should be noted that in their experiment Fitousi and colleagues [30] asked participants to discriminate only between angry and neutral faces and did not investigate the effects on other emotions. In addition to featural and configural characteristics of the stimuli, emotion perception can be affected by personal characteristics of the observer. Many studies have found that alexithymia, or difficulty in identifying emotions [31], can predict ability in the perception of facial expressions [32]. Experimental findings showed an association between high levels in alexithymia and atypical attentional processing of faces or atypical fixation patterns in face scanning [33,34]. In a recent fMRI study made by Rosenberg and colleagues (2020), it was found that subjects with high alexithymia levels have less sensitivity in the detection of anger signals in a seen face in the context of an emotional priming task. The authors concluded that difficulty in automatic perception of emotion can contribute to interpersonal relationship problems associated with alexithymia [35].

The aim of the present study was to test how occlusion due to a face mask can affect specific emotion recognition and to clarify the contribution of upper and lower face parts in emotion recognition and their role in the activation of featural or configural processing mechanisms. We tested subjects in a facial emotion recognition task (FERT) in which stimuli were presented in four different conditions characterized by different manipulations of the presented face. In masked face condition (MF) faces wore a surgical mask; in non-masked condition (NM) faces were fully visible; in eyes-only condition (EO) faces were cropped in order to make visible only the eyes region; in mouth-only condition (MO) faces were cropped to make only the mouth region visible. The rationale behind the presentation of these conditions is that configural processing seems to be automatic, so when a masked face is presented, it is possible that subjects attempt to process the face via configural mechanisms that are impaired by the visual occlusion due to the surgical mask. EO and MO conditions were presented to verify if a pure featural presentation of the faces could have different effects in terms of speed and accuracy due to the activation of featural processing and to evaluate the specific contribution of the different face features in emotion recognition.

Another aim was to assess if individual differences in alexithymia could relate to performance in different experimental conditions in order to clarify the relation with this personality trait and the process underlying emotion recognition. 

## 2. Materials and Methods

### 2.1. Participants

Thirty-one healthy Caucasian voluntary participants (16 males aged 21–58) with normal or corrected-to-normal visual acuity took part in the experiment. 

All participants had no history of neuropsychiatric disorders, as confirmed by clinical history and an anamnestic interview. Demographic information about our sample is listed in Table 1. Participants who took part in the experiment were treated following the Helsinki Declaration principles and provided informed consent prior to the experiment.

### 2.2. Procedure

Participants were asked to perform a facial emotion recognition task (FERT) in which they had to press a key to indicate the emotion expressed by faces of male and female Caucasian adults. Stimulus faces consisted of the portrait photos of 8 Caucasian adults (four men, four women) extracted from the NimStim Face Stimulus Set [36] expressing anger, happiness, sadness, and neutral expression. Each portrait was presented in the same facial expression 3 times for a total of 96 trials (8 faces × 4 emotions × 3 repetitions) in each experimental block. Four alternative sets of pictures were presented, one for each face manipulation condition: Not Masked (NM) consisting of a fully visible face; Masked Face (MF) in which a surgical mask was superimposed on the stimulus face; Mouth Only (MO) consisting of a cropped version of the stimulus to make only the lower part of the face visible; Eyes Only (EO) consisting of a cropped version of the stimulus to make only the upper part of the face visible (see Figure 1). 

The experimental procedure consisted of four blocks; only one set of stimuli was presented in each block. The order of the blocks was counterbalanced between participants. Stimulus presentation, timing, and data collection were controlled via the E-Prime (Psychology Software Tools, Pittsburgh, PA, USA) software, running on a laptop computer. Reaction times (RTs) and error rate (percentage of incorrect responses) were recorded.

Each block was preceded by a practice phase consisting of 10 trials, which were discarded from the analysis. Each trial started with a fixation cross at the center of the screen, which was replaced by the stimulus face after 1000 ms of fixation. Labels with emotions were placed in the lower part of the screen, in correspondence with the respective response keys, to facilitate the correct association of keys to the respective emotions; labels remained visible on the screen during the whole experimental procedure. 

The stimulus remained on the screen until a response was given. No feedback was given in the case of wrong responses. 

In order to assess the possible effects of participants’ mood, we administered four visual analogue scales (VAS). Each VAS consisted of a horizontal line, 10 cm in length, anchored by word descriptors at each end. 

The patient marked on the line the point that they felt represented their perception of their current emotional state. The VAS score was calculated by measuring in millimeters from the left-hand end of the line to the point that the patient marked. We used a VAS for happiness: 0 = unhappy and 10 = happy; for sadness: 0 = no sadness and 10 = sadness; for anger: 0 = no anger and 10 = anger; for mood: 0 = the worst mood ever and 10 = the best mood ever.

In order to assess participants’ perception of task difficulty we administered a visual analogue scale (VAS) for each experimental condition administered. Participants marked on the line the point they felt represented the difficulty of each experimental condition (NM, MF, MO, EO) administered. 

VAS on perceived difficulty had the same scale and evaluation method used for the VAS for mood assessment. 

Then the participants completed the Toronto Alexithymia Scale—20 (TAS-20) [37] questionnaire. In order to exclude subjects with a deficit in emotion recognition we administered the Comprehensive Affect Test System (CATS) [38]. The whole procedure lasted about 45 min. 

### 2.3. Statistical Analyses 

All subjects involved in the experiment showed normal scores in CATS, and no one was excluded due to impaired emotion recognition process. No outliers were present in the CATS scores (all scores were within ± 2 S.D. from the group mean). 

We used a 4 × 4 (4 face manipulation conditions × 4 face emotions) repeated measures ANOVA with reaction times for correct responses only and error rates (calculated as percentage of incorrect responses) as dependent variables. We made post hoc comparisons using paired sample t-tests with Bonferroni *p*-value correction for multiple comparisons (α = 0.012).

We correlated reaction times for correct responses and error rates with TAS-20 total score and TAS-20 subscales scores. 

VAS on perceived difficulty between face manipulation condition scores were entered in a repeated measure ANOVA with face manipulation condition as a within-factor. Post hoc comparisons were made using paired sample t-tests with Bonferroni *p*-value correction for multiple comparisons (α = 0.012).

## 3. Results

Error rates analysis showed the main effects of face manipulation condition (F(3,90) = 54.346, *p* < 0.001, η^2^ = 0.644), face emotion (F(3,90) = 64.404, *p* < 0.001, *η*^2^ = 0.682), and interaction between face manipulation conditions and face emotion (F(9,270) = 7.936, *p* < 0.001, *η*^2^ = 0.209).

To clarify the interaction between face manipulation conditions and face emotion, we carried out post hoc paired sample t-tests comparing each face manipulation condition with the NM condition.

Post hoc t-tests with Bonferroni *p*-value correction (*p* = 0.012) showed that participants made more errors in the MF than in the NM condition in sadness recognition, anger recognition, and neutral recognition. Differences in error rates between the MF and NM conditions in happiness recognition were not significant (see Figure 2). Detailed results are displayed in Table 2.

Comparing the EO condition with the NM condition, there were significant differences in sadness recognition, anger recognition, and neutral recognition, and no significant differences in happiness recognition (Table 2). Comparisons between the MO and NM conditions did not reach the significance threshold for any of the emotions shown. 

Reaction times analysis showed the main effects of face manipulation condition (F (3,90) = 29.298, *p* < 0.001, *η*^2^ = 0.491), face emotions (F (3,90) = 24.948, *p* < 0.001, *η*^2^ = 0.454), and the interaction between face manipulation conditions and face emotions (F (9,270) = 2.606, *p* = 0.007, *η*^2^ = 0.080). Post hoc t-tests with Bonferroni *p*-value correction (*p* = 0.012) showed that reaction times for correct responses were slower for all emotions in the MF condition compared to the NM condition (see Figure 3). Reaction times were also shorter in NM conditions than in EO conditions for all face emotion conditions (Table 3). 

No significant differences were found between MF and EO conditions or between NM and MO conditions (all *p* > 0.012). 

We calculated mean reaction times in each face manipulation condition regardless of the emotion. Results are shown in Table 4. Mean reaction times correlated positively with the TAS-20 score in the MO condition (*r* = 0.484, *p* = 0.006), in the NM condition (*r* = 0.477, *p* = 0.007), and in the EO condition (*r* = 0.374, *p* = 0.038). No correlations were found in the MF condition (*r* = 0.083, *p* = 0.656).

The TAS-20 subscale “difficulty in describing feelings” was correlated with reaction times in the MO (*r* = 0.491, *p* = 0.005) and NM *(r* = 0.530, *p* = 0.002) conditions. 

No significant correlations were found between the “externally oriented thinking” TAS -20 subscale and the reaction times. TAS-20 and relative subscales mean scores, standard deviation, and range are reported in Table 5.

VAS on perceived difficulty of face manipulation conditions results were entered in a repeated measure ANOVA with face manipulation condition as a within-factor; results show that participants perceived different degrees of difficulty between experimental conditions (F (3,90) = 47.116, *p* < 0.001, *η*^2^ = 0.611). Post hoc comparisons with paired sample t-tests showed that the NM condition was perceived as less difficult than other experimental conditions (NM vs. MF: 2.29 ± 1.465 vs. 5.94 ± 1.750, *p* < 0.001; NM vs. EO: 2.29 ± 1.465 vs. 5.77 ± 1.820, *p* < 0.001; NM vs. MO: 2.29 ± 1.465 vs. 3.77 ± 1.802, *p* < 0.001).

## 4. Discussion

The present study investigated how humans recognize emotions in faces wearing surgical masks. Typically, observers use information from different areas of the face in order to recognize the emotion expressed by the face they are looking at. In our experiment, participants performed a FERT observing fully visible faces, masked faces, eye regions or mouth regions of angry, happy, sad, or neutral faces.

Error rates analysis showed that participants made more errors in MF than in NM conditions for all emotions shown except happy faces. Our findings demonstrate that sadness and anger are the most misinterpreted emotions when only the eye region remains visible.

This result could be due to the fact that happiness recognition is less dependent on configural information than anger and sadness recognition [13]. Furthermore, the orientation of eyebrows is dramatically different between happiness, anger, or sadness, so it is possible that this unique visible feature enables people to better discriminate between these emotions in the context of the specific proposed task.

Another explanation could be that, in our set of stimuli, happiness was the only positive emotion and therefore could have been easily identified due to the marked difference in emotional valence.

Reaction times analysis showed that emotion recognition was slower in MF condition than in NM condition for all emotions expressed by the faces. This result confirms that wearing a face mask can impair emotion recognition due to the resulting perceptual occlusion of a relevant part of the face.

Furthermore, we found that for all emotions displayed, reaction times were shorter in NM condition than in EO condition; conversely, differences between NM and MO conditions were not significant. Previous studies have reported that despite the importance of the eye region, the mouth region plays a fundamental role in emotion recognition [22]. The importance of the mouth region in emotion recognition could be due to the fact that it is the part of the face with the most prominent facial movements and consequently provides more diagnostic features useful in emotion recognition.

Correlations between reaction times in different face presentation conditions and TAS-20 scores showed that participants with high TAS-20 scores were slower in emotion recognition in NM, EO, and MO conditions. The same correlation was not significant for the MF condition.

Subscales of TAS-20 “difficulty in identifying feeling” and “difficulty in describing feelings” refer to poor emotional awareness. Scores in these subscales correlated with reaction times in MO and NM conditions, while no correlation was observed between these subscales and EO and MF.

Alexithymia is a personality trait that consists in impaired capacity to identify and describe emotions [31]; additionally, individuals with high scores in alexithymia scale show impaired perception of facial expressions [32]. Participants with high scores in TAS-20 show atypical neurophysiological responses in components related to the attentional processing of faces [39] and tend to show atypical fixation patterns during face scanning [33,34].

It should be noted that, in the MF condition, stimuli had the shape of a whole face, so it is possible that in this condition participants scanned the stimuli in search of configural information, leading to an overall worsening of the performance, which may have hidden the specific alexithymia effects on reaction times in the FERT. In NM condition the possibility of scanning the whole face allowed participants with lower alexithymic traits to extract configural information more efficiently and therefore to perform better than participants with high alexithymia scores.

Our study is not free from limitations. In our sample, TAS-20 mean scores were slightly under the mean that can be found in the general population; thus, the variance in our sample could be restricted and therefore could reduce the magnitude of correlation, limiting the generalizability of our results to the general population. Another limitation is the absence of a group of alexithymic subjects; unfortunately, due to the small size of our sample it was not possible to create a control group composed of alexithymic subjects, which would have allowed us to compare alexithymic and non-alexithymic subjects and thus augment the clinical relevance of our findings.

Future studies with a larger sample size characterized by a high range in TAS-20 scores are needed to better estimate the strength of the correlation between alexithymic traits and emotion recognition.

## 5. Conclusions

The aim of the present study was to assess the effect of surgical masks on configural and featural processing of faces and subsequent effects on emotion perception. We found that surgical masks have a huge impact on emotion recognition and that the effect is dependent on the emotion showed by a seen face. More precisely, we found that surgical masks have a major impact in the recognition of angry and sad faces, while happiness recognition is less affected by the loss of visual information due to the perceptual occlusion produced by the surgical mask. Furthermore, we found correlations between alexithymia and reaction times in the recognition of emotion in non-masked faces, while no correlation was found in the masked face condition. This result suggests that in subjects with low levels of alexithymia, who should be more efficient in face scanning, the sight of a masked face can lead to the activation of configural face processing. Such activation is maladaptive in the case of masked faces because of the impossibility of accessing configural information; on the other hand, featural processing of the visible part of the face would be a more efficient strategy in terms of speed and accuracy.

While surgical masks are of fundamental importance to fight the spread of the pandemic, it should be noted that the impairment in emotion recognition can lead to communication problems. As noted by other authors [40], healthcare professionals must consider communication issues due to surgical masks in their everyday practice and find communication strategies to minimize the impact of the loss of perceptual information when interacting with patients [41].

From a clinical point of view, it is interesting to note that the differences we found in reaction time are similar to those found when comparing clinical populations with neurological or psychiatric disorders, characterized by a deficit in emotion recognition, with healthy controls [42,43,44]. These differences may have a strong impact on emotion perception, with huge consequences on the quality of interaction between individuals and related psychological wellbeing. Our results concerning a sample of healthy individuals show that surgical masks have a huge impact on emotion perception. These findings raise a question regarding the consequences that surgical masks can have in clinical populations affected by diseases involving emotion perception impairment such as autism spectrum disorder, major depressive disorder, and alcohol use disorder, where emotion perception deficit has a central role in the severity of symptoms and consequent social adaptation of patients [45].

Ideally, future investigations should also consider bodily aspects together with facial expressions [46]. Moreover, the use of morphed facial expressions could help in clarifying the differential effect of the eye and mouth regions in decoding emotions during real-life experience.

An interesting field of research could be the longitudinal monitoring of emotion recognition ability in participants living during the COVID-19 pandemic, in order to evaluate the possible adaptation of the human ability to identify emotions to a prolonged condition of reduced facial communication.

## Figures and Tables

**Figure 1 ijerph-19-02420-f001:**
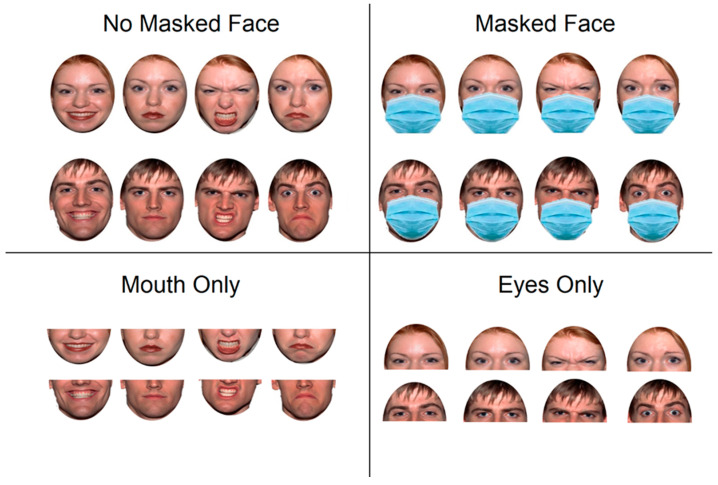
Experimental manipulation of the stimuli in each condition.

**Figure 2 ijerph-19-02420-f002:**
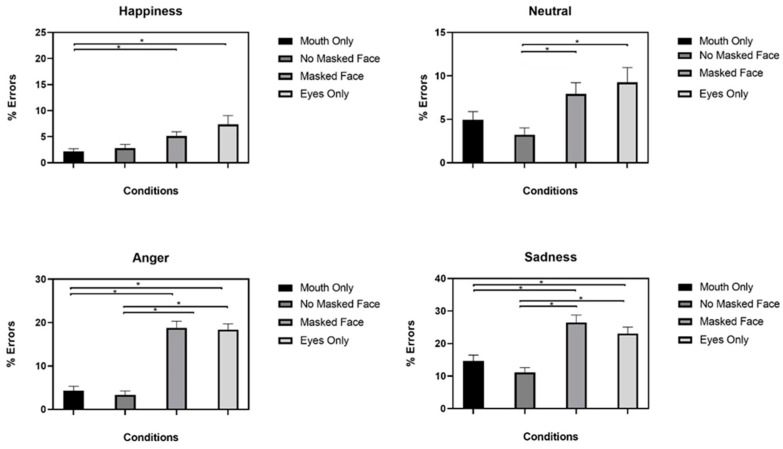
Error rates in face manipulation conditions for each emotion shown. * *p*-value ≤ 0.012.

**Figure 3 ijerph-19-02420-f003:**
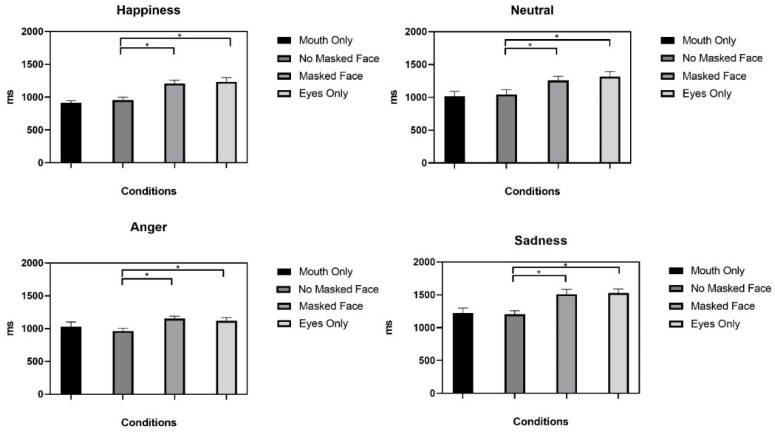
Mean reaction times for correct responses only in face manipulation conditions for each emotion shown. * *p*-value ≤ 0.012.

**Table 1 ijerph-19-02420-t001:** Demographic data for each participant.

Demographic Variable	Data
Sample Size	31 (16 M)
Age (Years; Mean ± SD)	32 ± 11
Education (Years; Mean ± SD)	17 ± 4
TAS- Score (Mean ± SD)	39.5 ± 9.37

**Table 2 ijerph-19-02420-t002:** Post hoc comparisons of error rates.

Emotions	Face Manipulation Conditions	Mean Difference	SD	*t*	df	Sig.
Happiness	NM 2.82 ± 3.94 MO 2.15 ± 3.01	0.67	5.05	0.74	30	0.465
Happiness	NM 2.82 ± 3.94 M 5.10 ± 4.65	−2.28	5.14	−2.47	30	0.019
Happiness	NM 2.82 ± 3.94 EO 7.39 ± 9.36	−4.56	10.50	−2.42	30	0.022
Neutral	NM 3.22 ± 4.39 MO 4.97 ± 5.09	−1.74	6.86	−1.42	30	0.167
Neutral	NM 3.22 ± 4.39 M 7.93 ± 7.16	−4.70	8.17	−3.20	30	0.003 *
Neutral	NM 3.22 ± 4.39 EO 9.27 ± 9.36	−6.04	9.18	−3.67	30	0.001 *
Anger	NM 3.36 ± 4.86 MO 4.30 ± 5.74	−0.94	6.15	−0.85	30	0.401
Anger	NM 3.36 ± 4.86 M 18.81 ± 8.32	−15.45	8.41	−10.23	30	<0.001 *
Anger	NM 3.36 ± 4.86 EO 18.41 ± 7.27	−15.05	7.19	−11.66	30	<0.001 *
Sadness	NM 11.15 ± 8.22 MO 14.65 ± 10.08	−3.49	9.19	−2.12	30	0.043
Sadness	NM 11.15 ± 8.22 M 26.48 ± 12.57	−15.32	15.86	−5.38	30	<0.001 *
Sadness	NM 11.15 ± 8.22 EO 23.11 ± 10.90	−11.96	14.01	−4.75	30	<0.001 *

Conditions are listed as NM = No Masked, MO = Mouth Only; EO = Eyes Only; M = Masked. * *p*-value ≤ 0.012 adjusted for Bonferroni correction for multiple comparisons.

**Table 3 ijerph-19-02420-t003:** Post hoc comparisons of reaction times for correct responses only.

Emotions	Face Manipulation Conditions (ms; Mean ± SD)	Mean Difference	SD	*t*	df	Sig.
Happiness	NM 957.33 ± 243.10 MO 916.41 ± 167.18	40.92	188.19	1.21	30	0.235
Happiness	NM 957.33 ± 243.10 M 1209.19 ± 281.68	−251.86	243.84	−5.75	30	<0.001 *
Happiness	NM 957.33 ± 243.10 EO 1233.80 ±367.66	−276.46	364.50	−4.23	30	<0.001 *
Neutral	NM 1042.88 ± 385.10 MO1016.86 ± 419.07	26.02	196.80	0.74	30	0.467
Neutral	NM 1042.88 ± 385.10 M 1258.00 ± 351.75	−215.12	298.40	−4.01	30	<0.001 *
Neutral	NM 1042.88 ± 385.10 EO 1313.94 ± 444.31	−271.06	317.21	−4.76	30	<0.001 *
Anger	NM 964.94 ± 223.95 MO 1031.59 ± 390.74	−66.64	256.85	−1.45	30	0.159
Anger	NM 964.94 ± 223.95 M 1151.74 ± 219.49	−186.80	178.61	−5.82	30	<0.001 *
Anger	NM 964.94 ± 223.95 EO 1118.97 ± 266.11	−154.03	193.98	−4.42	30	<0.001 *
Sadness	NM 1203.64 ± 304.12 MO 1223.72 ± 419.99	−20.07	249.88	−0.45	30	0.658
Sadness	NM 1203.64 ± 304.12 M 1511.57 ± 400.47	−307.92	385.49	−4.45	30	<0.001 *
Sadness	NM 1203.64 ± 304.12 EO 1527.12 ± 346.59	−323.47	366.80	−4.91	30	<0.001 *

Conditions are listed as NM = No Masked, MO = Mouth Only; EO = Eyes Only; M = Masked. * *p*-value ≤ 0.012 adjusted for Bonferroni correction for multiple comparisons. Asterisks denote statistically significant results.

**Table 4 ijerph-19-02420-t004:** Correlation between TAS-20 score and mean reaction times for correct responses only in different face manipulation conditions.

Face Manipulation Conditions		Difficulty Describing Feelings	Difficulty Identifying Feelings	Externally-Oriented Thinking	TAS-20 Total
MO	Pearson r	0.49 **	0.38 *	0.16	0.48 **
	*p*-value	0.005	0.034	0.383	0.006
EO	Pearson r	0.24	0.35	0.18	0.37 *
	*p*-value	0.199	0.056	0.326	0.038
NM	Pearson r	0.53 **	0.36 *	0.09	0.48 **
	*p*-value	0.002	0.050	0.608	0.007
MF	Pearson r	0.07	0.07	0.08	0.08
	*p*-value	0.705	0.726	0.661	0.656

Conditions are listed as NM = non-masked; MO = mouth only; EO = eyes only; M = masked. * = *p*-value < 0.05; ** = *p*-value < 0.01.

**Table 5 ijerph-19-02420-t005:** TAS-20; Externally-Oriented Thinking; Difficulty Identifying Feeling; Externally-Oriented Thinking Mean scores, standard deviations, and range.

	Mean	S.D.	Range
TAS-20	39.5	9.4	26–67
Difficulty Describing Feelings	11.1	3.8	5–23
Difficulty Identifying Feeling	13.5	5.0	7–25
Externally-Oriented Thinking	16.4	3.0	13–27

## Data Availability

Data not provided in the article must be shared at the request of other investigators for purposes of replicating procedures and results.

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
