# Peer review of "The Effect of Surgical Masks on the Featural and Configural Processing of Emotions"

_ijerph, 2022, doi:10.3390/ijerph19042420_

Round 1
Reviewer 1 Report
Maiorana et al. present a study on 31 healthy subjects evaluating the impact of surgical masks on emotion recognition using the FACS system. Their study design included pictures of faces with eyes or mouth only. They controlled for alexithymia.
The presented study methods and results seem to be scientifically sound and contribute to a very active field of research within the last 2 years. Findings are not essentially new or unexpected, but worthwhile publishing in a partially heated debate.
I have a few recommendations:
- Alexithymia is measured by the TAS-20, which is scientific standard. The authors present a Mean score of 39.5. It should be mentioned as a limitation that this mean total score is relatively low compared to the average score found in the general population. The variance with such a low mean score could be restricted and reduce the magnitude of correlations! Any interpretation of study results in correlation to this low TAS-score should therefore be done extremely cautiously or not at all. They should also not only give SD but also the range of TAS-Score in their cohort.
- The authors describe correlations of Subscores of the TAS, but I can’t find baseline data on that. At least the Mean (SD) and Range should be stated.
- The authors mention the CATS in the methods section, but no further mentioning. Why? Some kind of statement should be made in the results section, even if non-significant or inconclusive. Or delete it at all.
- line 198 (Fig2) showed -> shown
- line 216 (Fig 3) showed -> shown
- line 238 shows -> show
- line 240 all the others experimental conditions -> all other experimental conditions
- Discussion: The authors jump between present tense and past tense. E.g. line 249: “Error rates analysis shows….” vs line 261 “Reaction times analysis showed…”
- Lines 306-308: some kind of wording mistake in this sentence.
- I understand that the field is highly active and therefore literature search is immediately outdated once done. My literature search on this topic yielded several new articles only from the last few weeks, which could not be integrated into this manuscript. Nevertheless, among others, I came across the following two articles, that might be worthwhile for the authors to consider as references to put their work in the bigger frame of ongoing research. I leave this upon the authors discretion: Rosenberg et al. 2020 (doi: 10.1186/s12868-020-00572-6) and Marini et al 2021 (10.1038/s41598-021-84806-5)
Author Response
Maiorana et al. present a study on 31 healthy subjects evaluating the impact of surgical masks on emotion recognition using the FACS system. Their study design included pictures of faces with eyes or mouth only. They controlled for alexithymia.
The presented study methods and results seem to be scientifically sound and contribute to a very active field of research within the last 2 years. Findings are not essentially new or unexpected, but worthwhile publishing in a partially heated debate.
I have a few recommendations:
- Alexithymia is measured by the TAS-20, which is scientific standard. The authors present a Mean score of 39.5. It should be mentioned as a limitation that this mean total score is relatively low compared to the average score found in the general population. The variance with such a low mean score could be restricted and reduce the magnitude of correlations! Any interpretation of study results in correlation to this low TAS-score should therefore be done extremely cautiously or not at all. They should also not only give SD but also the range of TAS-Score in their cohort.
We thank the reviewer for the suggestion. We have added limitations about the generalizability of our results in the discussion section (lines 346 to 349). We have also added table 5 with mean, standard deviation and range for TAS-20 total score and subascales.
- The authors describe correlations of Subscores of the TAS, but I can’t find baseline data on that. At least the Mean (SD) and Range should be stated.
We thank the reviewer for the suggestion, We have added table 5 with mean, standard deviation and range for TAS-20 total score and subscales.
- The authors mention the CATS in the methods section, but no further mentioning. Why? Some kind of statement should be made in the results section, even if non-significant or inconclusive. Or delete it at all.
We thank the reviewer for the comment. We administered CATS to verify that no subjects had a deficit in emotion recognition so that we could have excluded subjects who deviated by ± 2 standard deviations from the mean. No subject was excluded due to altered emotion recognition performance measured by CATS. We have added an explanation of the rationale of CATS' administration in the methods section and the result of the administration in the statistical analyses section. Lines 199-200 and lines 204-205
- line 198 (Fig2) showed -> shown
We thank the reviewer for the suggestion. We have amended the manuscript
- line 216 (Fig 3) showed -> shown
We thank the reviewer for the suggestion. We have amended the manuscript
- line 238 shows -> show
We thank the reviewer for the suggestion. We have amended the manuscript
- line 240 all the others experimental conditions -> all other experimental conditions
We thank the reviewer for the suggestion. We have amended the manuscript
- Discussion: The authors jump between present tense and past tense. E.g. line 249: “Error rates analysis shows….” vs line 261 “Reaction times analysis showed…”
We thank the reviewer for the suggestion. We have amended the manuscript
- Lines 306-308: some kind of wording mistake in this sentence.
We thank the reviewer for the suggestion. We have amended the manuscript
- I understand that the field is highly active and therefore literature search is immediately outdated once done. My literature search on this topic yielded several new articles only from the last few weeks, which could not be integrated into this manuscript. Nevertheless, among others, I came across the following two articles, that might be worthwhile for the authors to consider as references to put their work in the bigger frame of ongoing research. I leave this upon the authors discretion: Rosenberg et al. 2020 (doi: 10.1186/s12868-020-00572-6) and Marini et al 2021 (10.1038/s41598-021-84806-5)
We thank the reviewer for the suggested references, we have added in the introduction sections of the manuscript lines 103-107 and lines 122-127
Reviewer 2 Report
Manuscript ID: ijerph-1596414
Manuscript title: The effect of surgical masks on featural and configural processing of emotions
This is a cross sectional study report with the title “The effect of surgical masks on featural and configural processing of emotions”. The authors had two aims to be confirmed by tests, as following:
- To test how occlusion due to face mask can affect specific emotion recognition and to clarify the contribution of upper and lower face parts in emotion recognition and their role in the activation of featural or configural processing mechanisms.
- To assess if individual differences in alexithymia could relate with performance in different experimental conditions in order to clarify the relation with this personality trait and the process underlying emotion recognition.
However, we have common knowledge that face masks would impair face recognition, in terms of increasing errors and decreasing speed of recognition (Grundmann, F. et al). Moreover, what is the clinical implication of error in face recognition or decreased speed, especially if the difference seems to be counted in milliseconds? In daily life, even if it takes longer to confirm people’s emotions, there is no clinically significant difference and one could confirm these emotions by additional observation or further interaction. There are also many factors that influence the image of people, such as recent stress, long-term mood, many different personality traits, and cultural expression of emotion. The authors demonstrate that sadness and anger are the more misinterpreted emotion compared to happiness when only the eyes region remains visible. It is better that the authors could elaborate more new clinical implication for the readers during COVID-19 pandemic.
The reviewer has the following main questions about this manuscript.
- The definitions of “featural ” or “configural” face cognition process are difficult to be understood and were not elaborated in the manuscript. These two processes were both impaired by mask according to the following statement of the authors, “Featural processing concerns the processing of information carried by specific face parts, for example the shape of the nose or the size of the mouth, while configural processing concern the spatial relation between face parts, for example the distance between eyes and mouth”. How to differentiate or define the occlusion of these two processing by masked face look? Are these two occlusions useful when people wear masks if they want to hide their anger or sadness? Please elaborate more on how face masks may influence featural or configural face cognition processes or any clinical or practical application.
- Could the authors describe the difference between the experimental face manipulation conditions: Masked Face (MF) and Eyes Only (EO)? These two groups shared similar characteristic features.
- It is rare to see people in daily situations showing mouth only like the Mouth Only (MO) picture. In what situations would this have clinical application? Is it like cases where the person is wearing sunglasses or maybe even masks covering the person’s eyes? And if so, what clinical significance does this imply?
- Regarding alexithymia as categorized by the Toronto Alexithymia Scale as used in the manuscript, many patients with psychiatric diagnosis share characteristic features with alexithymia, such as autism spectrum disorders (ASD), chronic dysthymia, post-traumatic stress disorder, anorexia nervosa, bulimia, major depression, panic disorder, social phobia, substance abusers. Although the manuscript mentioned that “All participants had no history of neuropsychiatric disorders,” how were these participants excluded? All participants with neuropsychiatric disorders should be excluded by psychiatrist, especially those who scored over 60 on the Toronto Alexithymia Scale.
- Alexithymia is a personality trait characterized by the subclinical inability to identify and describe emotions experienced by oneself. The core characteristic of alexithymia is marked dysfunction in emotional awareness, social attachment, and interpersonal relation. According to definition of alexithymia, it is reasonable that the authors found correlations between alexithymia and reaction times in the recognition of emotion in no masked face, while no correlation was found in the masked face condition. Thus, what is the significance in creating the hypothesis to assess if alexithymia could relate with facial emotion recognition?
- The Toronto Alexithymia Scale is a screening measure of deficiency in understanding, processing, or describing emotions. Do any of the participants meet the criteria of alexithymia? If any participants met the criteria, they should be used as a comparison group for clinical application. If no participants met the criteria, does this have any clinical significance if all participants only had differences on a self-reported screening test? The number of cases could be increased if no clinical significance was noted, especially if no comparison group was used.
- Please clearly explain the result: “This result suggests that also in subjects with low levels of alexithymia –who should be more efficient in face scanning– the vision of a masked whole face can lead to activation of configural face processing that is maladaptive in the case of masked face where featural processing is more efficient in terms of speed and accuracy.” Should the conclusion be that emotions are more easily recognized by looking at features individually, such as looking at mouth feature for happy emotions? If so, this should be more clearly stated.
- Please add a section for strengths and limitations for this manuscript.
Grundmann, F., K. Epstude and S. Scheibe (2021). "Face masks reduce emotion-recognition accuracy and perceived closeness." PLoS One 16(4): e0249792.
Author Response
Manuscript ID: ijerph-1596414
Manuscript title: The effect of surgical masks on featural and configural processing of emotions
This is a cross sectional study report with the title “The effect of surgical masks on featural and configural processing of emotions”. The authors had two aims to be confirmed by tests, as following:
- To test how occlusion due to face mask can affect specific emotion recognition and to clarify the contribution of upper and lower face parts in emotion recognition and their role in the activation of featural or configural processing mechanisms.
- To assess if individual differences in alexithymia could relate with performance in different experimental conditions in order to clarify the relation with this personality trait and the process underlying emotion recognition.
However, we have common knowledge that face masks would impair face recognition, in terms of increasing errors and decreasing speed of recognition (Grundmann, F. et al). Moreover, what is the clinical implication of error in face recognition or decreased speed, especially if the difference seems to be counted in milliseconds? In daily life, even if it takes longer to confirm people’s emotions, there is no clinically significant difference and one could confirm these emotions by additional observation or further interaction. There are also many factors that influence the image of people, such as recent stress, long-term mood, many different personality traits, and cultural expression of emotion. The authors demonstrate that sadness and anger are the more misinterpreted emotion compared to happiness when only the eyes region remains visible. It is better that the authors could elaborate more new clinical implication for the readers during COVID-19 pandemic.
We thank the reviewer for the comment. It is true that many factors influence the image of people as the reviewer noted; nevertheless decoding emotions expressed by a seen face is at the basis of human interactions and can have a huge impact on the quality of relationships between individuals. We believe that clarifying this aspect may be of strong interest, especially in healthcare where clinicians should consider the impact of surgical masks on emotion perception in the context of the relationship with patients. We have added this consideration in the conclusion section (lines 376-378)
The reviewer has the following main questions about this manuscript.
- The definitions of “featural ” or “configural” face cognition process are difficult to be understood and were not elaborated in the manuscript. These two processes were both impaired by mask according to the following statement of the authors, “Featural processing concerns the processing of information carried by specific face parts, for example the shape of the nose or the size of the mouth, while configural processing concern the spatial relation between face parts, for example the distance between eyes and mouth”. How to differentiate or define the occlusion of these two processing by masked face look? Are these two occlusions useful when people wear masks if they want to hide their anger or sadness? Please elaborate more on how face masks may influence featural or configural face cognition processes or any clinical or practical application.
We thank the reviewer for the comment. We have added an explanation of the role of surgical mask on featural and configural processing as defined in the paper. While in the process of a completely visible face, it is possible to rely on featural and configural information, masked face perception of emotions relies only on featural processing of the upper face region. Therefore surgical masks make impossible to process configural information and at the same time forces individuals to process only featural characteristics of the upper part of the face, without the possibility to access featural information of the lower part of a seen face.
As remarked by Ziccardi and colleagues (Ziccardi et al., 2022) healthcare workers should consider the difficulty in emotion recognition due to surgical mask in their professional activities. We have added explanation of the rationale of surgical mask occlusion (lines 53-60). We also added insight for clinical practice in the conclusions section (lines 374-376)
Ziccardi, S., Crescenzo, F., & Calabrese, M. (2022). “What Is Hidden behind the Mask?” Facial Emotion Recognition at the Time of COVID-19 Pandemic in Cognitively Normal Multiple Sclerosis Patients. Diagnostics, 12(1). https://doi.org/10.3390/diagnostics12010047
- Could the authors describe the difference between the experimental face manipulation conditions: Masked Face (MF) and Eyes Only (EO)? These two groups shared similar characteristic features.
We thank the reviewer for the comment. We have added an explanation of the rationale behind the stimuli manipulation in the introduction section (lines 130-142)
- It is rare to see people in daily situations showing mouth only like the Mouth Only (MO) picture. In what situations would this have clinical application? Is it like cases where the person is wearing sunglasses or maybe even masks covering the person’s eyes? And if so, what clinical significance does this imply?
We thank the reviewer for the comment. We have added an explanation of the rationale behind the stimuli manipulation in the introduction section (lines 130-142), MO condition was included to evaluate the specific contribution of the mouth region in the recognition of specific emotions via featural processing
Regarding alexithymia as categorized by the Toronto Alexithymia Scale as used in the manuscript, many patients with psychiatric diagnosis share characteristic features with alexithymia, such as autism spectrum disorders (ASD), chronic dysthymia, post-traumatic stress disorder, anorexia nervosa, bulimia, major depression, panic disorder, social phobia, substance abusers. Although the manuscript mentioned that “All participants had no history of neuropsychiatric disorders,” how were these participants excluded? All participants with neuropsychiatric disorders should be excluded by psychiatrist, especially those who scored over 60 on the Toronto Alexithymia Scale.
We thank the reviewer for the comment. All subjects were asked if they had a history of psychiatric disorder by the psychologist who carried out the experiment. We added this information in the material and methods section. (lines 150-151)
- Alexithymia is a personality trait characterized by the subclinical inability to identify and describe emotions experienced by oneself. The core characteristic of alexithymia is marked dysfunction in emotional awareness, social attachment, and interpersonal relation. According to definition of alexithymia, it is reasonable that the authors found correlations between alexithymia and reaction times in the recognition of emotion in no masked face, while no correlation was found in the masked face condition. Thus, what is the significance in creating the hypothesis to assess if alexithymia could relate with facial emotion recognition?
We thank the reviewer for the comment, to date it is unclear the link between alexithymia and automatic mechanism of emotion perception of a seen face, the rationale behind the hypothesis was to test if different perception mechanism could be activated in relation to alexithymia levels of the participants. We have added an explanation on the link between alexithymia and automatic processing of emotions and its clinical relevance in the introduction section lines (122-126)
- The Toronto Alexithymia Scale is a screening measure of deficiency in understanding, processing, or describing emotions. Do any of the participants meet the criteria of alexithymia? If any participants met the criteria, they should be used as a comparison group for clinical application. If no participants met the criteria, does this have any clinical significance if all participants only had differences on a self-reported screening test? The number of cases could be increased if no clinical significance was noted, especially if no comparison group was used.
We thank the reviewer for the comment, due to the small sample size of our sample it was not possible to create a control group based on alexithymia levels, we have added this limit in the discussion section lines (346-351)
- Please clearly explain the result: “This result suggests that also in subjects with low levels of alexithymia –who should be more efficient in face scanning– the vision of a masked whole face can lead to activation of configural face processing that is maladaptive in the case of masked face where featural processing is more efficient in terms of speed and accuracy.” Should the conclusion be that emotions are more easily recognized by looking at features individually, such as looking at mouth feature for happy emotions? If so, this should be more clearly stated.
We thank the reviewer for the comment, we have changed the text to make it clear
- Please add a section for strengths and limitations for this manuscript.
We have added limitations in the discussion section lines (346-351)
Grundmann, F., K. Epstude and S. Scheibe (2021). "Face masks reduce emotion-recognition accuracy and perceived closeness." PLoS One 16(4): e0249792.
Round 2
Reviewer 2 Report
The author made great effort to explain that decoding emotions expressed by a masked face can have an impact on the quality of relationships between individuals, especially in those with high alexithymia scores. The changes also clarified the definitions of featural and configural processes and how these processes relate to face emotion recognition when wearing a mask and the clinical significance. The authors described that surgical masks make it impossible to process configural information and at the same time force individuals to process only featural characteristics of the upper part of the face, without the ability to access featural information from the lower part of the face. Featural information may be more significant to people when identifying happy faces than faces with the other three emotions used in this manuscript. The authors clarified that surgical mask, EO and MO conditions were presented to verify if a pure featural presentation of the faces could have different effects in terms of speed and accuracy and to evaluate the specific contribution of the different face features in emotion recognition. The limitations and future considerations of the study were more clearly stated, especially in regards to alexithymia. Practical clinical application of the results seems to be limited, but the results can be taken into consideration for clinicians.
However, the authors did not respond to the question of whether there is any clinical implication of error in face recognition or decreased speed, especially if the difference seems to be counted in milliseconds, and at most less than 2 seconds, as shown in Table 3.
The reviewer appreciated that the authors had also added more references to highlight the clinical practice in this manuscript.
Author Response
The author made great effort to explain that decoding emotions expressed by a masked face can have an impact on the quality of relationships between individuals, especially in those with high alexithymia scores. The changes also clarified the definitions of featural and configural processes and how these processes relate to face emotion recognition when wearing a mask and the clinical significance. The authors described that surgical masks make it impossible to process configural information and at the same time force individuals to process only featural characteristics of the upper part of the face, without the ability to access featural information from the lower part of the face. Featural information may be more significant to people when identifying happy faces than faces with the other three emotions used in this manuscript. The authors clarified that surgical mask, EO and MO conditions were presented to verify if a pure featural presentation of the faces could have different effects in terms of speed and accuracy and to evaluate the specific contribution of the different face features in emotion recognition. The limitations and future considerations of the study were more clearly stated, especially in regards to alexithymia. Practical clinical application of the results seems to be limited, but the results can be taken into consideration for clinicians.
However, the authors did not respond to the question of whether there is any clinical implication of error in face recognition or decreased speed, especially if the difference seems to be counted in milliseconds, and at most less than 2 seconds, as shown in Table 3.
The reviewer appreciated that the authors had also added more references to highlight the clinical practice in this manuscript.
We thank the reviewer for his comment. Small differences in reaction times can have a huge impact on subsequent information processing. We have highlighted in the conclusions section how the differences in our study are similar to those found when comparing clinical populations with healthy controls in emotion perception tasks and the related clinical implication (lines: 379-390).